# Purification, Characterization, Prebiotic Preparations and Antioxidant Activity of Oligosaccharides from Mulberries

**DOI:** 10.3390/molecules24122329

**Published:** 2019-06-25

**Authors:** Erna Li, Shiyuan Yang, Yuxiao Zou, Weiwei Cheng, Bing Li, Tenggen Hu, Qian Li, Weifei Wang, Sentai Liao, Daorui Pang

**Affiliations:** 1Sericultural & Agri-Food Research Institute, Guangdong Academy of Agricultural Sciences/Key Laboratory of Functional Foods, Ministry of Agriculture and Rural Affairs/Guangdong Key Laboratory of Agricultural Products Processing, Guangzhou 510610, China; snoopylen@126.com (E.L.); YangSY0623@163.com (S.Y.); hutenggen@gdaas.cn (T.H.); liq@gdaas.cn (Q.L.); wangweifei@gdaas.cn (W.W.); liaost@163.com (S.L.); daorui66@163.com (D.P.); 2College of Food Science, South China Agricultural University, Guangzhou 510642, China; 3College of Food and Bioengineering, Henan University of Science and Technology, Luoyang 471000, China; 379062396@163.com; 4Guangdong Province Key Laboratory for Green Processing of Natural Products and Product Safety, Guangzhou 510641, China; bli@scut.edu.cn

**Keywords:** oligosaccharide, mulberry, chemical composition, prebiotics, antioxidant

## Abstract

A water-soluble oligosaccharide termed EMOS-1a was prepared by enzymatic hydrolysis of polysaccharides purified from mulberries by column chromatography. The chemical structure of the purified fraction was investigated by ultraviolet spectroscopy, Fourier-transform infrared spectroscopy, and gas chromatography–mass spectrometry, which indicated that galactose was the main constituent of EMOS-1a. Chemical analyses showed that the uronic acid and sulfate content of EMOS-1a were 5.6% and 8.35%, respectively, while gel permeation chromatography showed that EMOS-1a had an average molecular weight of 987 Da. The antioxidant activities of EMOS-1a were next investigated, and EMOS-1a exhibited concentration-dependent 1,1-diphenyl-2-picrylhydrazyl radical scavenging activity, Trolox equivalent antioxidant capacity, and ferric reducing antioxidant power. The level of proliferation of *Lactobacillus rhamnosus* reached 1420 ± 16% when 4% (*w/v*) EMOS-1a was added, where the number of colonies in MRS (de Man, Rogosa, and Sharpe) medium with no added oligosaccharide was defined as 100% proliferation. These results indicate that the oligosaccharide EMOS-1a could be used as a natural antioxidant in prebiotic preparations.

## 1. Introduction

Mulberry (*Morus nigra* L.) is widely cultivated in Asia, Africa, Europe, and America [1], and China has a long history of planting this tree [2]. In China, mulberries and other plants are used as a traditional medicine for treating fever, sore throat, anemia, and liver damage [3,4,5]. The therapeutic benefit of mulberries may be attributable to a variety of active ingredients, including polysaccharides, polyphenols, anthocyanins, and flavonoids [6,7,8].

Mulberry polysaccharides are macromolecular active substances with antioxidant, anti-fatigue, and immune-enhancing properties [9,10]. Prior research has indicated that the components obtained following deproteinization, decolorization, and desalting of mulberry polysaccharides have hypoglycemic activity in vitro [11]. In addition, mulberry polysaccharides have been shown to promote the proliferation and inhibit the apoptosis of spleen cells in vitro [12]. Chen et al. found that mulberry polysaccharides can regulate intestinal microbes by increasing the level of Bacteroidetes and decreasing that of Firmicutes [13].

Many studies have shown that plant polysaccharides become small-molecule oligosaccharides through physical, chemical, and enzymatic hydrolysis [14,15,16]. Bioavailability is thereby improved, and such oligosaccharides can improve probiotic activity [17]. We have previously found that mulberry oligosaccharides prepared by enzymatic hydrolysis have superior prebiotic effects compared with mulberry polysaccharides, isomaltooligosaccharides (IMOs), or galactooligosaccharides (GOSs) [18]. Wang found that enzymatically prepared rapeseed oligosaccharides significantly promoted the proliferation of *Bifidobacteria* and *Lactobacilli*, and the proliferation effect was superior to that induced by rapeseed polysaccharide [19].

In this study, an oligosaccharide termed EMOS-1a was purified from mulberries and characterized by chemical methods, and its in vitro antioxidant activity was also studied. Our findings provide a theoretical basis for the future development of EMOS-1a as a prebiotic product with antioxidant properties.

## 2. Results and Discussion

### 2.1. Isolation and Purification of Oligosaccharides

After hot water extraction, ethanol precipitation, hydrolysis by β-mannanase, and lyophilization, EMOS was obtained. EMOS was dissolved in distilled water and then separated by DEAE-52 cellulose column chromatography. Three fractions, EMOS-1, EMOS-2, and EMOS-3, were obtained (Figure 1A).

Based on previous studies, mulberry oligosaccharides prepared using β-mannanase can significantly promote the proliferation of *L. rhamnosus* [18]. Figure 1B shows data on the proliferation of *L. rhamnosus* cultured respectively with EMOS, EMOS-1, EMOS-2, and EMOS-3. Commercial preparations of the prebiotics IMO and GOS were used as positive controls. EMOS had a growth-promoting effect on *L. rhamnosus*, which reached a maximum proliferation of 824 ± 12% when EMOS was added at 4% (*w/v*). Note that 100% was defined as the growth observed in MRS medium with no added oligosaccharide. Compared with EMOS, EMOS-1 had a significantly superior proliferative effect on *L. rhamnosus* (*p* < 0.05). The proliferation of *L. rhamnosus* initially increased with increasing EMOS-1 concentration. When 4% (*w/v*) EMOS-1 was added, the proliferation level reached 1420 ± 16%. Both EMOS and EMOS-1 were superior to the positive controls, 5% (*w/v*) IMO (276 ± 9%), and 5% (*w/v*) GOS (395 ± 7%). However, the proliferation of *L. rhamnosus* was not significantly increased by 1–5% (*w/v*) EMOS-2 or EMOS-3. Upon addition of >4% (*w/v*) EMOS-1, the proliferation level decreased, which may be due to an increase in the osmotic pressure of the solution at high oligosaccharide concentration, which would cause the bacteria to dehydrate. 

These data demonstrate that the prebiotic effect of the oligosaccharide, EMOS-1, was concentration-dependent. The degree of polymerization in the carbohydrate structure affects the proliferation of probiotic bacteria. In general, carbohydrates with a low degree of polymerization are more readily available to probiotic bacteria [20]. We have previously found that mulberry oligosaccharides with a lower degree of polymerization have a greater proliferative effect than mulberry polysaccharides [18]. The chemical structure, monosaccharide composition, degree of branching of the carbohydrate chains, and solubility in water also modulate the prebiotic effects of oligosaccharides [21,22]; polysaccharides with a simple chemical structure, a large number of branches ending in carbohydrate chains, and high solubility in water are considered to have an enhanced beneficial effect on probiotic bacteria [23].

EMOS-1 was further purified by Sephadex G-100 chromatography. A fraction termed EMOS-1a was collected, corresponding to a single, sharp, symmetrical peak (Figure 1C). When 4% (*w/v*) EMOS-1a was added, the proliferation of *L. rhamnosus* reached the highest observed value of 1420 ± 16%.

### 2.2. Characterization of EMOS-1a

As a homogeneous oligosaccharide, EMOS-1a was chosen for further study. Chemical analysis showed that the uronic acid and sulfate content of EMOS-1a were 5.6% and 8.35%, respectively. The oligosaccharide had no UV absorption at 260 or 280 nm, indicating that EMOS-1a contained no protein or nucleotides (Figure 2A).

Polysaccharides with a triple-helix structure form a complex with the dye, Congo red [24]. Compared with Congo red, the maximum absorption wavelength of the complex shows a bathochromic shift, and the product is purple-red in color. When the added NaOH concentration exceeds a certain value, the maximum absorption wavelength drops sharply [25]. Structural assessment of EMOS-1a is shown in Figure 2B. The maximum absorbance of the complex formed by EMOS-1a and Congo red decreased only slightly with increasing NaOH concentration. This indicated that EMOS-1a did not have a triple-helix structure.

Molecular weight analysis by GPC is shown in Figure 3. The average molecular weight of EMOS-1a was determined to be 987 Da from the standard curve.

The monosaccharide composition of EMOS-1a was measured by GC–MS analysis. GC–MS analysis of seven standard monosaccharides is shown in Figure 4A. Based on the retention time and relative percentage of each sugar (Figure 4B), EMOS-1a was composed of galactose.

### 2.3. FT-IR Analysis

The FT-IR spectrum is commonly used to determine the functional groups in organic molecules. As shown in Figure 5, the spectrum of EMOS-1a displayed a broad and strong absorption peak at 3351 cm^−1^ (3500–3100 cm^−1^), which could be assigned to stretching vibrations of hydroxyl (–OH) groups in the polysaccharide and the water involved in hydrogen bonding. The absorption peak around 2929 cm^−1^ (3000–2800 cm^−1^) was attributed to the C–H stretching vibration of the methyl groups. These absorption peaks are characteristic of polysaccharides [26]. There was no absorption peak in the range 1740–1760 cm^−1^, indicating that EMOS-1a does not contain esterified carboxyl groups. Absorption peaks were detected in the region 1700–1300 cm^−1^, which indicated that the polysaccharide contained a carboxyl group. The absorption peak at 1639 cm^−1^ indicated that EMOS-1a contained uronic acid [27]. There was no absorption peak at 1541 cm^−1^, confirming the absence of protein. The absorption peak at 1400–1200 cm^−1^ was assigned to C–H bending, and the peak at 1242 cm^−1^ to the presence of a sulfate radical [28]. The stretching peak at 1079 cm^−1^ indicated the presence of a pyranose ring [29].

### 2.4. Antioxidant Activity

Antioxidant activities of EMOS-1a were determined based on DPPH radical scavenging activity and the FRAP and ABTS tests. The DPPH radical scavenging results are shown in Figure 6A. DPPH free radicals are stable and purple in color in organic solvents (λ_max_ = 517 nm). When an antioxidant is added, the DPPH radical is removed and the absorption at this wavelength is attenuated. The scavenging of DPPH free radicals by EMOS-1a was dependent on the EMOS-1a concentration. When the concentration of EMOS-1a was 600 μg/mL, the scavenging ability was 46.96 ± 1.68%. The scavenging effect was weaker than that of vitamin C, and the half-clearance concentration of EMOS-1a for DPPH free radicals was approximately 650 μg/mL. 

The results of the ABTS test showed a similar trend (Figure 6B). The Trolox-equivalent antioxidant capacity of EMOS-1a was positively correlated with its concentration. ABTS is oxidized to green ABTS^+^ under the action of an oxidizing agent, and the production of ABTS^+^ is inhibited in the presence of an antioxidant. The Trolox-equivalent antioxidant capacity of a sample can be determined by measuring the absorbance of ABTS^+^ at 734 nm. At concentrations of 100–800 μg/mL, the Trolox-equivalent antioxidant capacity of EMOS-1a was 0.6020 ± 0.0088 to 5.8420 ± 0.1155, which is lower than that of vitamin C.

The FRAP value of EMOS-1a was concentration-dependent (Figure 6C). In acidic conditions, an antioxidant can reduce Fe^3+^-TPTZ to produce blue Fe^2+^-TPTZ. The FRAP value in the sample can then be obtained by measuring the absorbance of Fe^2+^-TPTZ at 593 nm. At concentrations of 100–800 μg/mL, the FRAP value of EMOS-1a was 0.0376 ± 0.0114 to 0.4627 ± 0.01015, which was lower than that of vitamin C.

## 3. Material and Methods

### 3.1. Material and Chemicals

Mulberries were supplied by a farm in the Baiyun District, Guangzhou, Guangdong Province, China. MRS (de Man, Rogosa, and Sharpe) medium and agar powder were purchased from Guangdong Huankai Microbial Sci. & Tech. Co., Ltd. (Guangzhou, China). IMO, GOS, β-mannanase (50 U/mg), rhamnose, arabinose, and mannose were purchased from Shanghai Yuanye Bio-Technology Co., Ltd., (Shanghai, China). DEAE-52 cellulose and Sephadex G-100 were purchased from Shanghai Ryon Biological Technology Co., Ltd., (Shanghai, China) Ribose, xylose, and 1,1-diphenyl-2-picrylhydrazyl (DPPH) were purchased from Sigma-Aldrich (St. Louis, MO, USA). Fucose and galactose were purchased from Amresco (Solon, OH, USA). All other chemicals were of analytical grade and purchased from Guangzhou Chemical Reagent Factory (Guangzhou, China).

### 3.2. Bacterial Strain

*Lactobacillus rhamnosus* (ATCC7469) was purchased from the Guangdong Culture Collection Center. The strain was stored at −80 °C in MRS broth containing 25% (*v/v*) glycerol.

### 3.3. Preparation, Isolation, and Purification of Mulberry Oligosaccharides

Mulberry polysaccharides were prepared according to the method described by Chen et al. with some modifications [13]. Polysaccharides in the mulberry powder were extracted in a water bath at 80 °C for 4 h. The resulting extract was reduced to a quarter of its original volume by vacuum filtration, followed by precipitation for 24 h with four volumes of 95% (*v/v*) ethanol at 4 °C. The precipitate was collected by centrifugation (9000× *g* for 15 min), and the pellet was dissolved in water to obtain a crude mulberry polysaccharide solution. For the production of the enzymatic hydrolysate containing the mulberry oligosaccharides (EMOS), the crude mulberry polysaccharide solution was incubated with 5% (*w/v*) β-mannanase at 50 °C for 4 h and then lyophilized. The polysaccharide content of the mulberry polysaccharide solution and EMOS were detected with the phenol–sulfuric acid method using d-glucose as the standard [30].

EMOS was dissolved in distilled water (10% (*w/v*)) and then loaded onto a DEAE-52 cellulose column (2.5 cm × 25 cm) previously equilibrated with distilled water at room temperature. The column was eluted with distilled water and a step gradient of 0.1, 0.3, and 0.5 mol/L NaCl at a flow rate of 1 mL/min. The eluates were collected with an automatic fraction collector (10 mL per tube). The elution profile detected by the phenol–sulfuric acid assay showed three main elution peaks, namely EMOS-1, EMOS-2, and EMOS-3, which were then lyophilized. Fraction EMOS-1 was selected for further fractionation because of its superior proliferation effect (see Methods 2.4). Size-exclusion chromatography on a Sephadex G-100 column (2.5 cm × 25 cm) with distilled water at a flow rate of 1 mL/min yielded a fraction that was denoted EMOS-1a, which was also then lyophilized.

### 3.4. Effect of Oligosaccharides on the Growth of L. rhamnosus

*L. rhamnosus* (ATCC7469) was employed to investigate the effect of EMOS, EMOS-1, EMOS-1a, EMOS-2, and EMOS-3 on bacterial proliferation. IMO and GOS were used as positive controls. EMOS, EMOS-1, EMOS-1a, EMOS-2, EMOS-3, IMO, and GOS were respectively added to MRS medium (5 mL) at final concentrations of 1, 2, 3, 4, and 5% (*w/v*). The mixtures were then inoculated with 1% (*v/v*) overnight culture of *L. rhamnosus* and incubated with shaking at 180 rpm at 37 °C for 12 h. Samples were plated to count the number of resulting colonies. The number of colonies in MRS medium with no added oligosaccharide was defined as having a proliferation level of 100%, where the proliferation level was calculated as
(1)$$\mathrm{Proliferation}\;\mathrm{level}\;=\frac{\mathrm{The}\;\mathrm{number}\;\mathrm{of}\;\mathrm{colonies}\;\mathrm{in}\;\mathrm{MRS}\;\mathrm{medium}\;\mathrm{with}\;\mathrm{oligosaccharide}}{\mathrm{The}\;\mathrm{number}\;\mathrm{of}\;\mathrm{colonies}\;\mathrm{in}\;\mathrm{MRS}\;\mathrm{medium}\;\mathrm{with}\;\mathrm{no}\;\mathrm{oligosaccharide}}\times 100\%.$$

### 3.5. Chemical Characterization of EMOS-1a

#### 3.5.1. Determination of Chemical Composition

The uronic acid content of the EMOS fractions was estimated with the vitriol–carbazole method using d-galacturonic acid as a standard, and the sulfate content was tested with the barium chloride–gelatin method [31].

#### 3.5.2. UV Spectroscopy

Ultraviolet and visible (UV–vis) absorption spectra of the EMOS fractions were recorded with a Shimadzu UV-1800 spectrophotometer (190–500 nm, Kyoto, Japan).

#### 3.5.3. Determination of Structure

Congo red was used to determine whether EMOS-1a has a triple-helix structure, as described by Wood et al., based on a shift in the maximum absorption wavelength [32]. Briefly, 0.1, 0.2, 0.3, 0.4, 0.5, 0.6, 0.7, 0.8, 0.9, and 1.0 mol/L NaOH was mixed with 2 mg/mL EMOS-1a and 80 μmol/L Congo red at a volume ratio of 2:1:1, and the mixtures were analyzed with a Shimadzu UV-1800 spectrophotometer.

#### 3.5.4. Fourier-Transform Infrared (FT-IR) Spectroscopy

EMOS-1a (1 mg) was mixed with KBr powder and then pressed into pellets for analysis. The FT-IR spectrum was recorded in the range 400–4000 cm^−1^ on a Vertex 70 FT-IR spectrometer (Bruker Optics, Ettlingen, Germany).

#### 3.5.5. Molecular Weight Determination

The average molecular weight of EMOS-1a was determined by gel permeation chromatography (GPC), which was performed on three tandem columns (Acquity APC AQ 450, Acquity APC AQ 125, and Acquity APC AQ 45, Waters Corp., Milford, CT, USA). Standard dextran, including ZAP-DXT180 (molecular mass, 180 Da), ZAP-DXT350 (342 Da), ZAP-DXT500 (505 Da), ZAP-DXT1K (1000 Da), ZAP-DXT3K (2800 Da), and ZAP-DXT4K (3400 Da), were used as molecular mass markers.

#### 3.5.6. Analysis of Monosaccharide Composition

The monosaccharide composition of EMOS-1a was analyzed by gas chromatography–mass spectrometry (GC–MS) [33] with some modifications. Briefly, EMOS-1a (10 mg) was hydrolyzed to monosaccharides using trifluoroacetic acid (2 mL, 4 mol/L) at 110 °C for 6 h. The hydrolyzed solution was evaporated to dryness, added to hydroxylamine hydrochloride (10 mg) and pyridine (1 mL) for reaction for 30 min at 90 °C, and then acetylated with acetic anhydride (1 mL) for 30 min at 90 °C. Thus, the monosaccharides were derivatized as acetylated aldononitriles. The final product was analyzed by GC–MS using an Agilent 6890 GC instrument (Agilent Technologies Co., Ltd., Colorado Springs, CO, USA) equipped with an HP-1701 column and an Agilent 5973 MS detector. Nitrogen was used as the carrier gas (1 mL/min). The temperature of the column was initially set at 100 °C, then increased to 280 °C at 10 °C/min, followed by 280 °C for 15 min. The injection temperature was 280 °C and the temperature of the mass spectrometer ion source was 230 °C. Seven monosaccharides (rhamnose, ribose, fucose, arabinose, xylose, mannose, and galactose) were converted into their acetylated derivatives as standards to identify the composition of the oligosaccharides.

### 3.6. Antioxidant Activity of EMOS-1a In Vitro

#### 3.6.1. DPPH Radical Scavenging Assay

The DPPH assay followed the methodology developed by Brand-Williams et al. [34] with modifications. Briefly, 3 mL EMOS-1a aqueous solutions (100, 200, 400, 600, and 800 μg/mL) and 3 mL DPPH in ethanol (0.10 mmol/L) were mixed and shaken vigorously. The mixtures were kept in the dark for 30 min and the absorbance was then measured at 517 nm. The blank comprised water instead of EMOS-1a, and vitamin C was used as the positive control. The antioxidant power of the mixtures was calculated with the equation:(2)$$\mathrm{Scavenging}\;\mathrm{effect}\;\mathrm{of}\;\mathrm{DPPH}\;\mathrm{radical}\;\left(\%\right)=\frac{A_{\mathrm{blank}}-A_{\mathrm{sample}}}{A_{\mathrm{blank}}}\times 100$$

#### 3.6.2. ABTS [2,2′-Azino-bis(3-ethylbenzothiazoline-6-sulfonic) Acid] Assay

The ABTS radical-scavenging activity of EMOS-1a was determined with an ABTS assay kit (Beyotime Biotechnology Co., Ltd., Shanghai, China). ABTS working liquor was prepared by reacting ABTS stock solution with potassium persulfate solution at a 1:1 volume ratio. The mixture was stored at room temperature for 12–16 h in the dark before use and left to stabilize for 2 to 3 days. The concentration of the resulting blue-green ABTS radical solution was adjusted to give an absorbance of 0.7 ± 0.05 at 734 nm. A total of 200 µL of ABTS working solution and 10 µL EMOS-1a solution were mixed in 96-well plates, which were protected from light. Absorbance was measured at 734 nm after 2–6 min of incubation at room temperature. Antioxidant activity was expressed as the Trolox-equivalent antioxidant capacity of the EMOS-1a solution.

#### 3.6.3. FRAP (Ferric Reducing Antioxidant Power) Assay

The antioxidant activity of EMOS-1a was determined with an antioxidant capacity assay kit and the FRAP method (Beyotime Biotechnology Co., Ltd., Shanghai, China). A working solution was prepared by mixing a TPTZ dilution, TPTZ solution, and detection buffer at a ratio of 10:1:1 (*v/v*), and this FRAP working solution was then kept at 37 °C. Calibration solution, blank, or sample (5 μL) were added to 180 μL of working solution. Absorbance was measured at 593 nm after 3–5 min incubation at 37 °C. The FRAP values of the samples were calculated from a linear calibration curve and expressed as mmol FeSO_4_ equivalents.

### 3.7. Statistical Analysis

All data are presented as the mean ± standard deviation (SD) of three independent experiments. Statistical significance (*p* < 0.05) between treatments was analyzed by one-way analysis of variance, followed by Duncan’s multiple range test. All statistical analyses were performed with SPSS software version 19.0 (SPSS Inc., Chicago, IL, USA).

## 4. Conclusions

In this study, the oligosaccharide EMOS-1a was prepared by enzymatic hydrolysis of polysaccharides purified from mulberries by DEAE-52 cellulose and Sephadex G-100 column chromatography. EMOS-1a consisted of galactose with an average molecular weight of 987 Da. In addition, the antioxidant activity of EMOS-1a was positively correlated with its concentration. EMOS-1a promoted the proliferation of *L. rhamnosus*; when 4% (*w/v*) EMOS-1a was added, the proliferation level reached 1420 ± 16%, which indicated that EMOS-1a has potential as a natural antioxidant in prebiotic foods. This research provides a basis for further studies of mulberry prebiotics. The anti-digestive properties and proliferative effect of EMOS-1a on human fecal microbiota will be investigated in future studies.

## Figures and Tables

**Figure 1 molecules-24-02329-f001:**
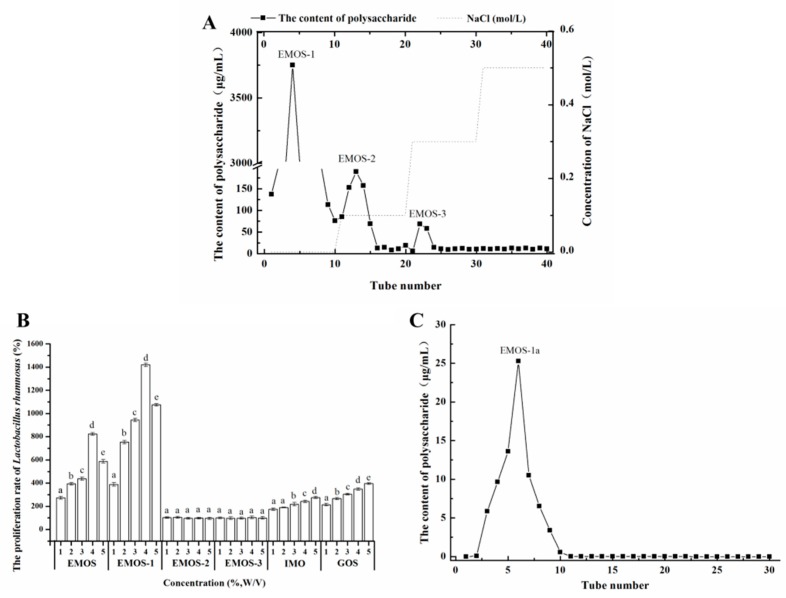
Elution profile of the enzymatic hydrolysate of mulberry oligosaccharides (EMOS) from a DEAE-52 cellulose column (**A**). Proliferation of *Lactobacillus rhamnosus* cultured for 12 h at 37 °C in medium containing different carbohydrates (**B**). Elution of EMOS-1a from a Sephadex G-100 chromatography column (**C**). Means with the same letter (for each carbohydrate added) are not significantly different (*p* > 0.05) according to Duncan’s multiple range test. Each experiment was repeated three times. Abbreviations: IMO, isomaltooligosaccharides; GOS, galactooligosaccharides; EMOS, enzymatic hydrolysate of mulberry oligosaccharides.

**Figure 2 molecules-24-02329-f002:**
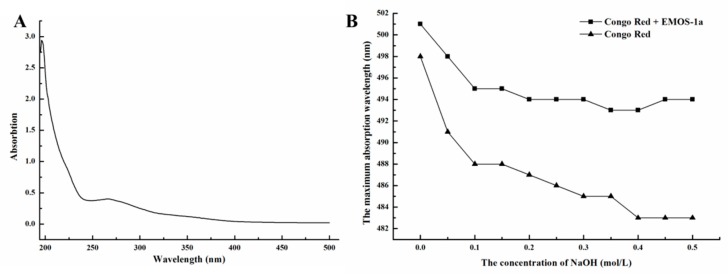
UV spectrum of EMOS-1a (**A**). Structural analysis of EMOS-1a (**B**); “Congo red” indicates that Congo red was mixed with NaOH, “Congo red + EMOS-1a” indicates that Congo red was treated with EMOS-1a before the addition of NaOH.

**Figure 3 molecules-24-02329-f003:**
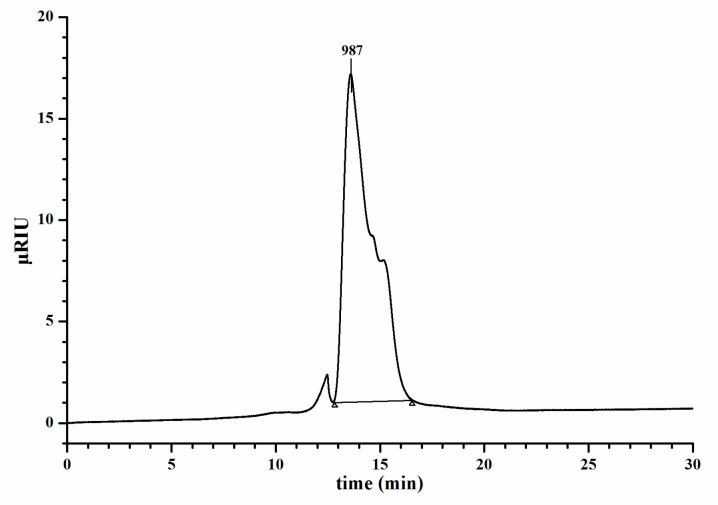
Molecular weight analysis of EMOS-1a by gel permeation chromatography.

**Figure 4 molecules-24-02329-f004:**
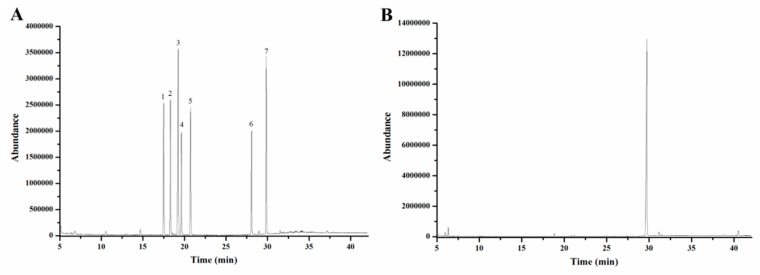
Monosaccharide composition analysis by gas chromatography–mass spectrometry of mixed monosaccharide standards (**A**). 1: Rhamnose, 2: ribose, 3: fucose, 4: arabinose, 5: xylose, 6: mannose, 7: galactose. Monosaccharide composition analysis of EMOS-1a (**B**).

**Figure 5 molecules-24-02329-f005:**
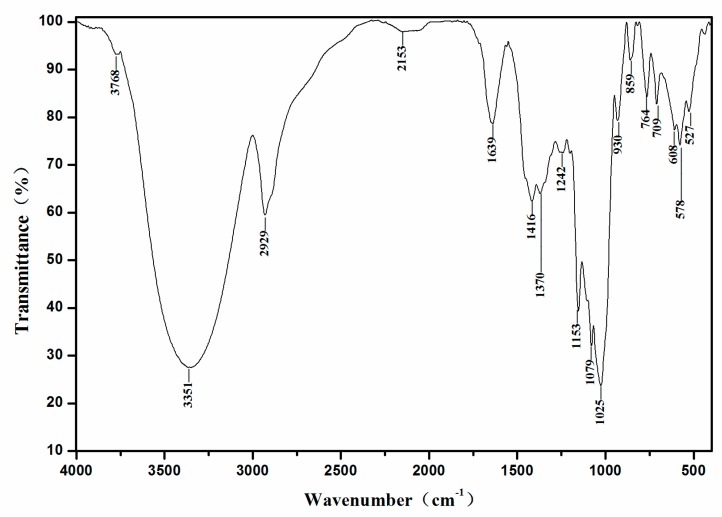
Fourier-transform infrared spectrum of EMOS-1a.

**Figure 6 molecules-24-02329-f006:**
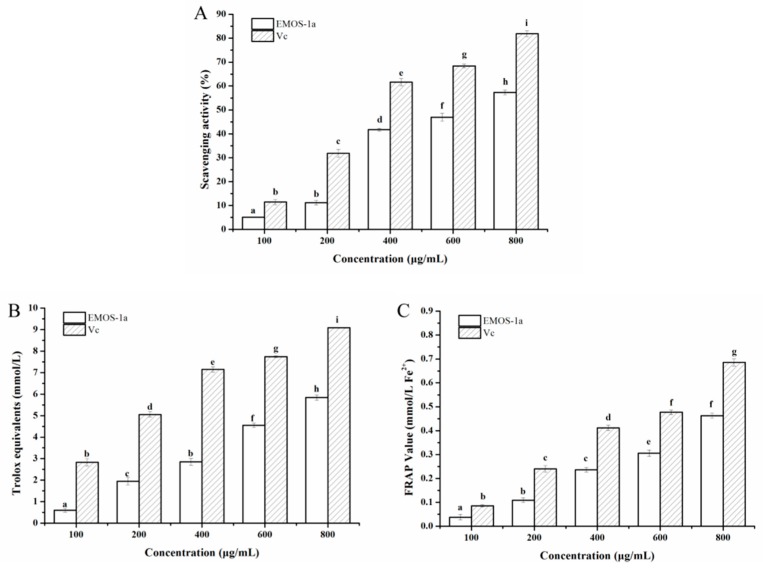
Antioxidant activity of EMOS-1a. 1,1-diphenyl-2-picrylhydrazyl (DPPH) radical scavenging effect (**A**), ferric-reducing antioxidant power (**B**), and 2,2′-azino-bis(3-ethylbenzothiazoline-6-sulfonic) acid (ABTS) radical-scavenging activity (**C**). Means with the same lowercase letter are not significantly different (*p* > 0.05) according to Duncan’s multiple range test. Each experiment was repeated three times and presented as the mean ± standard deviation (SD). Vitamin C (Vc) was used as the positive control.

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
