# Peer review of "Purification, Characterization, Prebiotic Preparations and Antioxidant Activity of Oligosaccharides from Mulberries"

_molecules, 2019, doi:10.3390/molecules24122329_

Round 1

Reviewer 1 Report

The paper presented by Erna Li and co-workers shows the preparation  of a water-soluble oligosaccharide termed EMOS-1, by enzymatic hydrolysis of polysaccharides purified from mulberries by column chromatography.

The obtained results revealed the obtained fraction could be used as a natural antioxidant in prebiotic preparations. Presentation of experimental data showing the isolation, purification and characterization of oligosaccharides is very important.

This work  brings new knowledge to the subject, taking into account the multitude of articles that refer to the extraction, analysis and biological studies of mulberries. The manuscript is easy and clear to read.

The presented paper may be publish in the Molecules after minor revision.

Some abbreviations are not explained for example Vc; in  figure 6 ( section 2) who is a, b,c, etc

Author Response

Some abbreviations are not explained for example Vc; in  figure 6 ( section 2) who is a, b,c, etc

Response: Thank you for your suggestion. We have added more details in page 6, line 165-168 in our revised manuscript.

Reviewer 2 Report

An interesting and well designed paper.

Author Response

An interesting and well designed paper.

Response: Thank you for your comments.

Reviewer 3 Report

This manuscript discusses the chemical and biological characteristics of polysaccharide extracts from mulberries. The manuscript is solid, well done and of good quality. I recommend it for publication.

Nonetheless, the following issues need to be addressed:

1)      Given that the characterization of these extracts is aimed at their use in prebiotic preparations, in my opinion the title should consider this aspect;

2)      Regarding figure 6, since they are the result of three different experiments, the authors should explain why SD and / or SEM are not indicated;

3)      Line 32, modify the sentence “In China, mulberries and other plants are used as a traditional, medicine for treating fever, sore throat, anemia, and liver damage” and insert the references: Bonini SA et al., Cannabis sativa: A comprehensive ethnopharmacological review of a medicinal plant with a long history. J Ethnopharmacol. 2018 Dec 5;227:300-315; Kumar A et al., Cannabimimetic plants: are they new cannabinoidergic modulators? Planta. 2019 Jun;249(6):1681-1694.

Author Response

1) Given that the characterization of these extracts is aimed at their use in prebiotic preparations, in my opinion the title should consider this aspect;

Response: Thank you for your suggestion. We have amended the title.

2) Regarding figure 6, since they are the result of three different experiments, the authors should explain why SD and / or SEM are not indicated;

Response: Thank you for your suggestion. We have added more details in page 6, line 165-168 in our revised manuscript.

3) Line 32, modify the sentence “In China, mulberries and other plants are used as a traditional, medicine for treating fever, sore throat, anemia, and liver damage” and insert the references: Bonini SA et al., Cannabis sativa: A comprehensive ethnopharmacological review of a medicinal plant with a long history. J Ethnopharmacol. 2018 Dec 5;227:300-315; Kumar A et al., Cannabimimetic plants: are they new cannabinoidergic modulators? Planta. 2019 Jun;249(6):1681-1694.

Response: Thank you for your suggestion. We have added more details in page 1, line 33-34 in our revised manuscript.